# Comparative genomics reveals LINE-1 recombination with diverse RNAs

## Graphical abstract

## Authors

Cheuk-Ting Law, Kathleen H. Burns

## Correspondence

cheuk-ting_law@dfci.harvard.edu (C.-T.L.),
kathleenh_burns@dfci.harvard.edu (K.H.B.)

## In brief

Using comparative genomics across mammals, Law and Burns uncover hundreds of LINE-1 insertions fused with diverse host RNAs, including previously unrecognized chimeras with *Alu*, small RNA, and mRNA sequences. These findings reveal recurrent RNA and LINE-1 recombination mechanisms that potentially drive transposon diversification and innovation throughout mammalian evolution.

## Highlights

- Developed TiMEstamp, a pipeline to systematically date TE insertions from MSA

- Compiled a large LINE-1 chimera catalog with over 700 events

- Discovered new LINE-1 chimeric partners (i.e., small RNAs, *Alu* elements, and mRNAs)

- Transcriptionally inactive LINE-1 loci can hijack external promoters to retrotranspose

Law & Burns, 2026, Cell Genomics 6, 101165
May 13, 2026 © 2026 The Author(s). Published by Elsevier Inc.

CellPress

## Article

# Comparative genomics reveals LINE-1 recombination with diverse RNAs

Cheuk-Ting Law[1,2,3,*] and Kathleen H. Burns[1,2,3,4,*]
[1]Department of Pathology, Dana-Farber Cancer Institute, Boston, MA 02115, USA
[2]Department of Pathology, Harvard Medical School, Boston, MA 02115, USA
[3]Broad Institute of MIT and Harvard, Cambridge, MA 02142, USA
[4]Lead contact
*Correspondence: cheuk-ting_law@dfci.harvard.edu (C.-T.L.), kathleenh_burns@dfci.harvard.edu (K.H.B.)

## SUMMARY

Long interspersed element-1 (LINE-1, L1) retrotransposons are the most abundant protein-coding transposable elements (TEs) in mammalian genomes and have shaped genome content over 170 million years of evolution. LINE-1 is self-propagating and mobilizes other sequences, including *Alu* elements. Occasionally, LINE-1 forms chimeric insertions with non-coding RNAs and mRNAs, but there are no comprehensive catalogs of LINE-1 chimeras. To address this, we developed timing mobile element insertions (TiMEstamp), a computational pipeline that leverages multiple sequence alignments (MSAs) to estimate the age of LINE-1 insertions and identify candidate chimeric insertions where an adjacent sequence arrives contemporaneously. With this pipeline, we discovered new chimeric insertions involving small RNAs, *Alu* elements, and mRNA fragments. Additionally, we saw evidence that LINE-1 loci with defunct promoters can acquire regulatory elements from nearby genes to restore expression and retrotransposition activity. These discoveries highlight the recombinatory potential of LINE-1 RNA with implications for genome evolution, TE domestication, and somatic retrotransposition.

## INTRODUCTION

Mammalian long interspersed element-1 (LINE-1, L1) retrotransposons are non-long terminal repeat (non-LTR) transposable elements (TEs) and represent the most abundant protein-coding, self-propagating retroelements in their hosts.[1–4] While LINE-1 protein-coding sequences are relatively conserved, their promoters vary in size (1–2 kb) and sequence across species.[1,5] Their persistent activity over more than 170 million years (Ma) of mammalian evolution has profoundly shaped genomes; approximately 30% of the human genome (1 billion bp) has been written by the LINE-1 reverse transcriptase (RT).[6–8]

Humans each inherit a complement of retrotransposition-competent LINE-1 loci transcribed by the internal CpG-rich RNA polymerase II promoter. LINE-1 RNA encodes two proteins: open reading frame 1 protein (ORF1p),[9,10] an RNA-binding protein, and ORF2p,[11–13] which possesses endonuclease (EN) and RT activities. During retrotransposition, these activities work in concert to execute target-primed reverse transcription (TPRT), using a genomic DNA nick to prime reverse transcription of the LINE-1 RNA to complementary DNA (cDNA). TPRT intermediates are resolved to yield double-stranded LINE-1 genomic insertions.[14–18] Insertions mediated by ORF2p possess several sequence hallmarks, including target-site duplications (TSDs),[19–21] which are short duplicated genomic sequences flanking both the 5′ and 3′ ends of the templated insertion; the

presence of an EN recognition motif (TT/AAAA, the slash indicates the cut site)[16,22,23] and a poly(A) tail at their 3′ end.[24,25]

Apart from copying its own sequence (in *cis*), LINE-1 ORF2p can also reverse transcribe and insert other sequences into the genome (in *trans*). The most notable examples in humans are non-coding short interspersed element (SINE) retrotransposons, including *Alu*[26] and composite "SINE-R, VNTR, and *Alu*" (SVA; VNTR is a variable number of tandem GC-rich repeats) elements.[27,28] By hijacking the TPRT mechanism, *Alu* and SVA have generated approximately 1.3 million and 6,000 copies in the human genome, respectively.[6] LINE-1 also drives the insertion of processed (spliced) messenger RNAs (mRNAs) in *trans*,[29,30] forming pseudogenes, and retrotransposes small non-coding RNAs, including transfer RNAs (tRNAs),[31] Y RNAs,[32] small nuclear RNAs (snRNAs),[33–35] small nucleolar RNAs (snoRNAs),[36] and ribosomal RNAs (rRNAs).[31] Collectively, hundreds of thousands of copies of these non-coding-RNA-derived sequences are present in the genome.

LINE-1 produces chimeric insertions with some RNA species, fusing short non-LINE-1 sequences with the 5′ end of full-length or 5′ truncated LINE-1, forming a bipartite insertion in a single event. To date, LINE-1 has been reported to form chimeric insertions with U1–U6 snRNAs[33–35] and 5S rRNA.[37] Among these, U6/LINE-1 chimeric insertions are the most frequent and best characterized; in these chimeras, the U6 segment is consistently full-length and occurs at the 5′ end of the LINE-1 and in the sense direction relative to the LINE-1.[35] Like other TPRT events, these

chimeras often occur at the LINE-1 EN motif (TT/AAAA) and are flanked by TSDs. The formation of U6/LINE-1 chimeric insertions may occur through two distinct mechanisms: RNA ligation, generating a chimeric RNA template for reverse transcription,[38] or template switching during TPRT, where the ORF2p RT shifts from the LINE-1 RNA to a nearby RNA molecule, concatenating both cDNAs.[35]

Building a comprehensive census of 3′-LINE-1 chimeras in the genome has been challenging. Candidate chimeras were primarily identified by searching for a "bait sequence" (a known sequence with a predilection for chimera formation, e.g., U6) in the genome and examining its proximity to LINE-1 elements,[33–35,37] followed by confirmation of TSDs and other TPRT signatures. Degradation of sequence homologies over time and independently occurring "nested" insertions of more recently active TEs can obscure key sequence signatures and pose challenges for recognizing chimeras. Furthermore, reliance on *a priori* knowledge of bait sequences also limits the ability of this approach to discover novel chimeric insertions.

To address these challenges, we developed a computational pipeline—timing mobile element insertions (TiMEstamp) from multiple sequence alignments—that applies a comparative genomics approach to estimate LINE-1 insertion time and identify chimeric LINE-1 insertions without reliance on bait sequences. First, we identify insertions encompassing LINE-1 and non-LINE-1 sequences that appear contemporaneously in evolutionary history using genome-wide multiple sequence alignment (MSA) of 464 genome assemblies across 438 mammalian species.[39] Next, we filter these candidates by searching for sequence hallmarks of LINE-1-mediated insertions. To overcome challenges posed by sequence divergence over time, we build consensus sequences of insertion alleles from MSA, an approach that greatly improves TSD detection.

Using this pipeline with the human genome assembly as a reference, we identified all known species of 5′-L1 chimeras, previously unrecognized RNA repeats, *Alu* elements, and mRNA transcripts that have formed LINE-1 chimeras over a broad time span in mammalian evolution. These findings indicate new mechanisms by which LINE-1 forms chimeric insertions, including *trans*-splicing events. Additionally, we see evidence that LINE-1 insertions with defunct promoters can acquire regulatory elements from nearby genes that restore their capacity for retrotransposition. These discoveries underscore the recombinatory potential between cellular RNAs and retroelements with implications for transposon evolution, TE domestication, and retrotransposon activity in tissues.

## RESULTS

### Inferring the occurrence of a retrotransposon insertion using MSA

Retrotransposon insertions are homoplasy free and have unambiguous directionality. In other words, insertion of a specific retroelement sequence at a specific site in the genome and its associated target site changes collectively represent unique features of a single mutagenic event. Seeing the same features in a different genome essentially always reflects identity by descent. Further, the ancestral or antedating allele is the "pre-insertion"

allele; the derived allele is the insertion allele. Thus, comparative genomics using MSA across species is a powerful tool to delineate features of retrotransposon insertions and infer their ages with respect to speciation.

We focused on studying human LINE-1 elements, which number over 500,000.[6] These include the oldest, L1M (including L1MA–E families, mammalian specific); L1P (including L1PA and L1PB, primate specific); and the youngest, L1HS (encompassing L1PA1, human specific). Sequence-homology-based analyses estimated that L1M families were active 60–150 million years ago (mya), L1P families 7–80 mya, and L1HS beginning ∼3 mya.[5,40,41] This prolonged activity has resulted in the significant accumulation of LINE-1 elements in the human genome.

To infer the timing of TE insertions with respect to speciation, we developed a computational pipeline—TiMEstamp—to analyze the MSA of 464 genome assemblies, computed by Hiller's group and accessible via the UCSC Genome Browser.[39,42,43] We transformed the alignment into a binary format, where regions were classified as either aligned (indicating the presence of the target sequence) or gapped (indicating its absence), without requiring nucleotide-level agreement. An alignment score of 1 indicated that the sequence (e.g., a LINE-1) was fully present in a species, and a score of 0 indicated its complete absence, with intermediate values reflecting deterioration of sequence homology over time or artifacts of alignment (Figure S1). To minimize noise from partial alignments in a particular genome assembly, we averaged alignment scores at the sister-clade level, where each clade shares a single common ancestor that diverged from the human lineage at the same branchpoint (Figure 1). Using these scores, we classified insertions as present or absent within each clade. We concluded by placing the arrival time of each insertion at the common ancestor of a monophyletic clade where the insertion is present in all members of that clade and absent in other taxa.

### Inferring the arrival of LINE-1 families in the human lineage

Phylogenetic analyses grouped these 438 species into 11 sister clades (or taxons) with respect to the human reference assembly, defining 12 distinct time points when LINE-1 elements can insert into the ancestral human genome (Figure 2A). We can appreciate, for example, that the primate-specific L1PA4 subfamily appeared subsequent to the divergence of apes and monkeys (corresponding to time point 5 in Figure 2A), while L1PA10 was introduced earlier, before the New and Old World monkey separation (at time point 7, Figures 2A and 2B). Mammalian L1M subfamilies antedate these. L1MA8 was present in the genomes of a common ancestor shared by primates and Glires (e.g., rodents and rabbits). In contrast, older LINE-1 elements, such as L1MB2, were shared in all species evaluated except Xenarthra and Afrotheria (e.g., moles and elephant) and Metatheria (e.g., kangaroo) clades or, in the case of L1ME4, except in Metatheria (Figure 2B).

To evaluate the accuracy of our approach, we tested these inferences against RepeatMasker annotations,[44] in which Smit et al. used 3′ sequence homology to infer the history of LINE-1 elements and annotate LINE-1 families.[41] Using our MSA-informed approach, we confirmed that L1P families are primate specific. Consistent with the original finding and other studies,[5,40,41] the

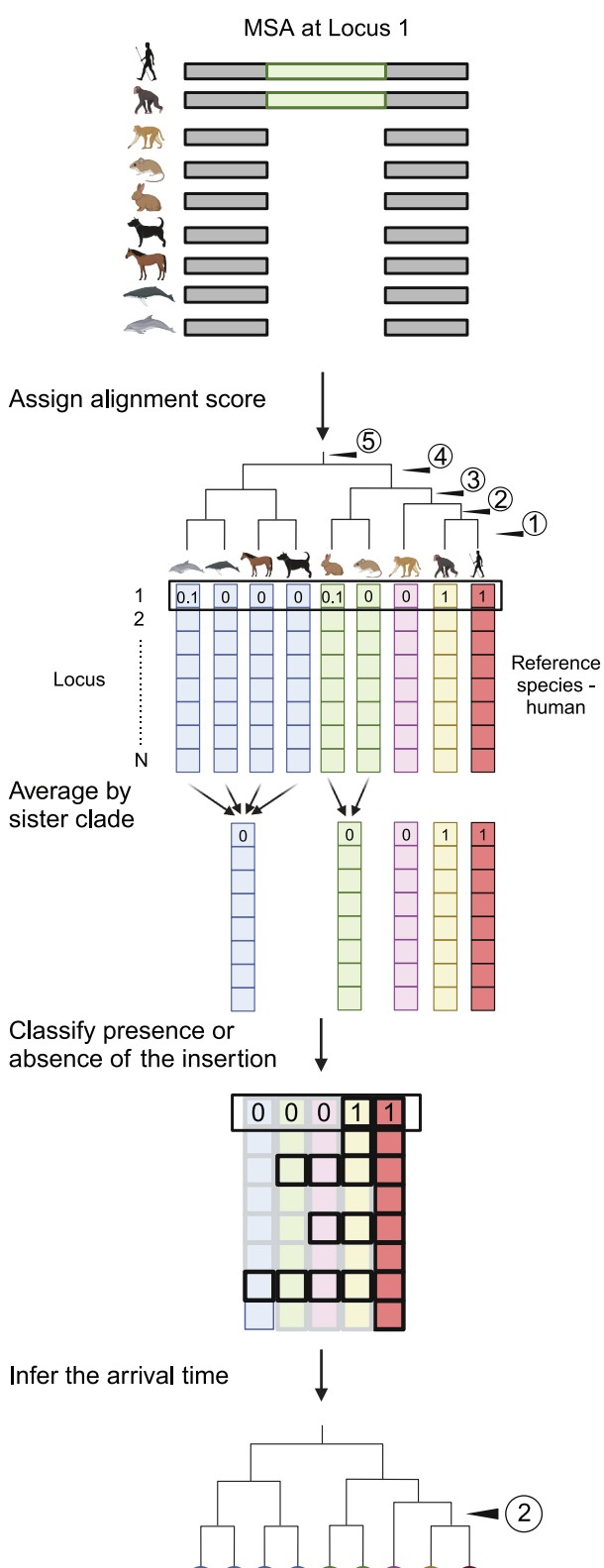

**Figure 1. TiMEstamp: A computational pipeline for inferring the evolutionary timing of sequence insertions**

Multiple sequence alignment (MSA) data are used relative to the human genome to define ancestral (pre-insertion) vs. insertion alleles. For each species, alignment to an insertion found in the human genome is scored 1 (complete alignment), signifying shared presence of the sequence; 0 (no alignment) represents a gap or absence of the sequence. Scores between 0 and 1 may be attributable to deterioration of sequence homology or alignment artifacts. To reduce noise, alignment scores are averaged at the sister-clade level, stepping away from the human reference. An insertion is inferred to have originated in a common ancestor of a monophyletic clade when it is present in all members of that clade/taxon and absent in other taxa/taxon.

youngest L1M families (L1MA6–L1MA1) are also primate specific (Table S1). Furthermore, this chronological ordering closely aligns with LINE-1 ages calculated by homology-based and de-fragmentation analyses[40] (Figure 2C). As examples, we illustrate sister-clade-based alignment scores for all loci of three representative subfamilies: the *Homo sapiens*-associated L1HS, the primate-specific L1PA5, and the broadly mammalian L1MA5 (Figure 2D). In each map, rows represent individual LINE-1 loci, while columns display the sister-clade-level averaged alignment scores, with the inferred insertion time point for each locus on the left. As expected, most annotated L1HS loci are human specific; however, 244 loci (18.1% of the L1HS loci annotated in RepeatMasker) also appear in other genome assemblies, indicating earlier insertion times. Of these, 176 loci (72.1%) were present in human and chimpanzee[45] or in human, chimpanzee, and gorilla[46] genome assemblies but absent from other assemblies, suggesting that these 176 loci would be better labeled as primate LINE-1. Notably, approximately 90% of these loci lack the 3′ UTR sequence and the 3′ 700 bp of ORF2 used for homology-based classification,[41] so these loci are challenging for RepeatMasker to accurately assign. For L1PA5, most loci were introduced at time point 5 (the divergence of apes and Old World monkeys), whereas L1MA5 displayed a broader activity interval, spanning time points 7 and 8, after the primates and Glires split. Collectively, these findings confirm the utility of MSA to recapitulate the known evolutionary succession of LINE-1 activity.

We expanded TiMEstamp analysis to all TEs annotated in RepeatMasker. Specifically, we analyzed all *Alu* loci in the human genome assembly, revealing a peak in retrotransposition activity at time point 7 and then a gradual decline, reaching low levels after time point 5 (Figure S2A). However, with the emergence of *Homo sapiens*, there appears to be a resurgence of *Alu* germline retrotransposition as previously reported.[47,48] In addition, analysis of SVA elements confirmed that these are hominid specific and evolutionarily young,[49,50] with most insertions occurring around time point 3 (Figure S2B), corresponding to the common ancestor of humans, chimpanzees, and gorillas.

Together, these findings underscore that TiMEstamp accurately reconstructs the insertion histories of diverse types of TEs and provides a reliable framework for studying TE dynamics on evolutionary timescales.

### Distinguishing sequential and chimeric LINE-1 insertions using MSA

MSA provides insight not only into when a DNA segment was introduced into the genome but also into whether two adjacent

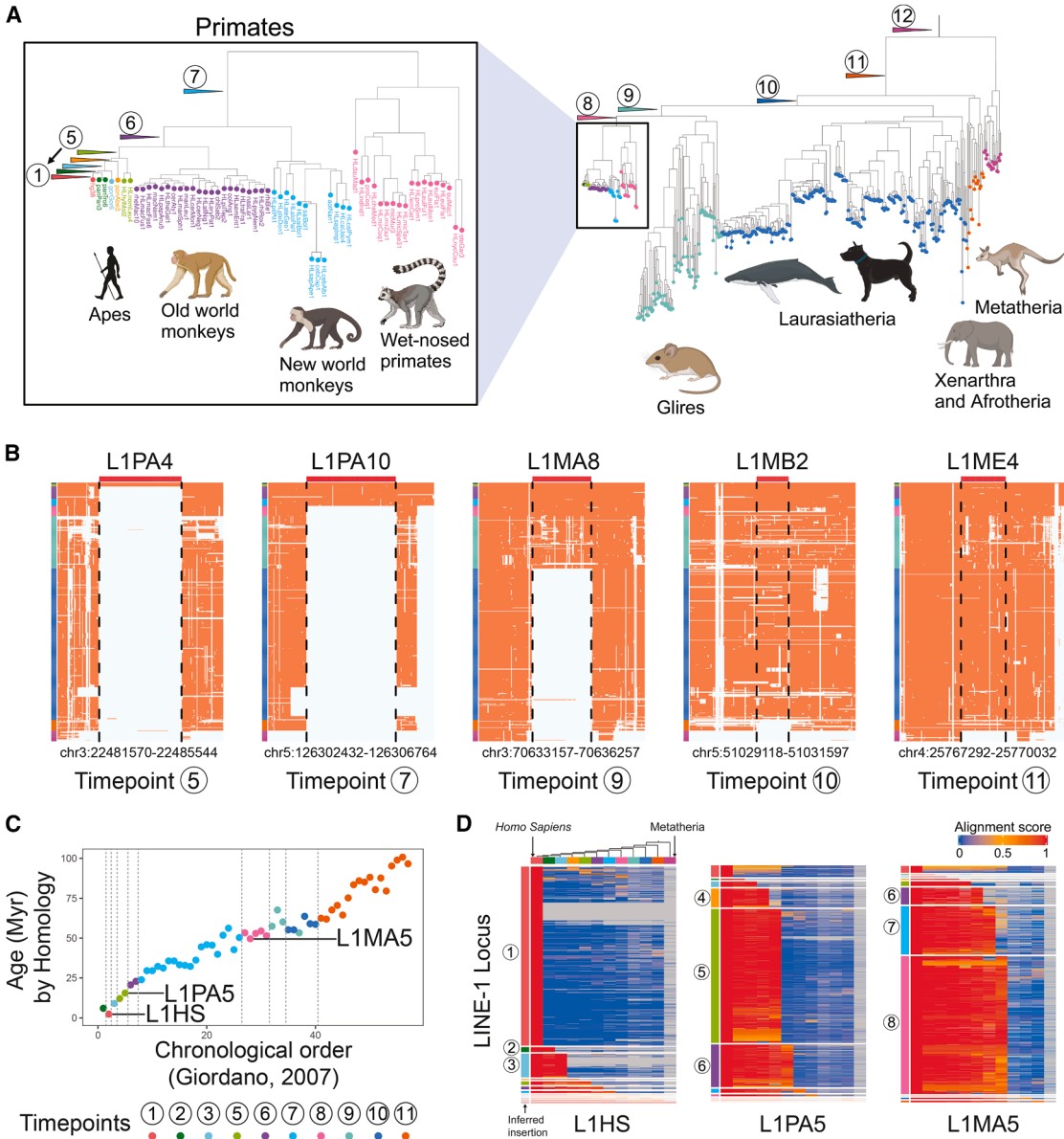

**Figure 2. Inferring LINE-1 insertion times in mammalian evolution**

(A) The phylogenetic tree, a phylogram of 464 genomes across 438 species, is organized into 11 sister clades/taxon and *Homo sapiens* (highlighted in different colors) and 12 ancestral time points. Species having more than one genome in the dataset are mostly from the Laurasiatheria clade.

(B) MSA profiles of 5 LINE-1 loci. Chromosomal locations are from the human genome assembly (hg38), and alignments of syntenic intervals in other species are arranged vertically. The color labels on the far left side of each profile match the sister clade from (A); orange in the MSA represents alignment, while gray indicates gaps or no alignment. LINE-1 subfamilies (L1PA4, L1PA10, L1MA8, L1MB2, and L1ME4) were annotated by RepeatMasker; their inferred insertion time points are noted below the MSA profile.

(C) Inferred insertion time points (dot colors), the chronological order of LINE-1 subfamilies calculated from defragmentation (*x* axis), and ages calculated using divergence from the subfamily consensus sequence (*y* axis) are highly correlated.

(D) Alignment scores against the human genome for three LINE-1 subfamilies (L1HS, L1PA5, and L1MA5, red indicates alignment). Each row represents a distinct locus in the subfamily; each column corresponds to the sister clade/taxon-level averaged alignment scores (far left column: *Homo sapiens*, far right column: Metatheria). The phylogenetic tree above illustrates the relative evolutionary relationship between sister clade/taxon. The inferred insertion time point for each locus is indicated on the left of the graph. Most L1HSs were inserted at time point 1; 244 loci (18.1%) appear to have earlier time points. Most L1PA5s were inserted at time points 5 and 6, while L1MA5 was primarily integrated at time points 7 and 8. This visualization highlights the successive activity intervals of these LINE-1 families.

insertions occurred sequentially or contemporaneously. Sequential insertions are inferred when one DNA segment is found in a group of species sharing a distant common ancestor, while its neighboring sequence appears restricted to a more recently related subgroup, indicating separate integration events, one following the other. In contrast, contemporaneous insertions are revealed when both DNA segments are consistently paired (present or absent together) across all species in the MSA and potentially integrated in a single molecular event (Figure 3A).

This approach allows us to identify chimeric LINE-1 insertions, where a LINE-1 retrotransposon integrates along with additional, unrelated sequences at its 5′ end in a single mutagenic event. U1, U2, U4, U5, and U6 snRNAs, U3 snoRNAs, and 5S rRNAs are all known to form chimeric insertions with LINE-1.[34,35,37,51] To begin, we compiled a catalog of chimeric insertions of these known types by manually inspecting MSA profiles of LINE-1 elements with these RNA sequences nearby. All together, we compiled a comprehensive catalog of 253 chimeric insertions (Table S2). An example MSA of a U6/LINE-1 chimera is shown in Figure 3B. Among these chimeras, 70.9% involved recombination between LINE-1 and U6 spliceosomal snRNA, 10.5% involved 5S rRNA, and 8.5% involved U3 snoRNA. Their relative abundance is consistent with prior studies.[34,35,37,38,51]

U6/LINE-1 chimeric insertions have occurred recurrently in human evolution; each appears as an independent recombination event joining a U6 and the 3′ end of a LINE-1 and usually does not further self-propagate after insertion. Most accumulated between time points 8 (post-primate/Glires divergence, ~90 mya) and 6 (post-monkey/ape divergence, ~30 mya) (Figures 2A and 3C). In these chimeras, the U6 portion is typically full-length (106 nt) (Figure 3D), while the LINE-1 portion is frequently 5′ truncated.[33,35,37] Interestingly, full-length U6 sequences are more abundant in U6/LINE-1 chimeric insertions compared to genomic U6 (Figure S3A), whereas the LINE-1 component shows more frequent 5′ truncation than genomic LINE-1 overall (Figure S3B). Consistent with prior studies,[33,37] almost all chimeric U6 (98.3%) are oriented in the same direction as their associated LINE-1 (sense orientation) (Figure 3D). These recurrent patterns indicate that a common mechanism underlies these chimeric insertions: an RNA ligation joining U6 to the LINE-1 3′ end before TPRT[38] or a LINE-1 RT template switch that concatenates these sequences. Similarly, U3 snoRNA sequences in U3/LINE-1 chimeric insertions exhibit a strong bias toward the sense orientation (95.5%). Unlike U6/LINE-1 chimeras, the U3 segments associated with LINE-1 are recurrently truncated (Figure S3C), which contrasts with the bimodal length distribution of genomic U3 sequences (Figure S3D). This suggests that a specific cleavage event produces the U3 fragments[52] that are then prone to recombination with LINE-1.

5S rRNA segments in 5S/LINE-1 chimeras are typically in antisense relative to their associated LINE-1 and truncated. Commonly, the start of the 5S rRNA is at the junction with the LINE-1 and extends until its truncation at positions 46 and 90 (Figure 3E), corresponding to loops C and D in 5S rRNA. The truncation at position 46 is more commonplace in these chimeric insertions than when all genomic 5S rRNAs are considered (Figure S3E), again suggesting the possibility of targeted

processing.[53,54] Of 27 5S/LINE-1 insertions, 20 show 7–17 bp TSDs flanking the insertion, and 14 occur at the LINE-1 EN motif (Table S2).

All U5 sequences observed in U5/LINE-1 chimeric insertions are truncated at their 3′ ends, and 80% are found in the antisense orientation relative to LINE-1.

These characteristics of U3, U5, and U6 chimeric LINE-1 insertions are consistent with previous reports.[33,35,37] Given our larger catalog of events, however, we can better see recurrent patterns for less common LINE-1 chimeras (e.g., length distributions, RNA repeat breakpoints, and orientation tendencies) and develop hypotheses for how these insertions arise.

### Novel RNA species that form chimeric LINE-1 insertions

To discover new types of chimeric LINE-1 insertions, we used TiMEstamp to systematically identify sequences immediately 5′ (upstream) of LINE-1 fragments inserted contemporaneously with the downstream LINE-1. These upstream sequences appear as gaps relative to the pre-insertion allele and were compared to existing annotations in the RepeatMasker database,[44] transcript annotation (Gencode),[55] long non-coding RNA (lncRNA) databases, RNAcentral,[56] and a human genome assembly.[6]

Using this pipeline, we recovered 76% of the previously manually curated chimeric LINE-1 insertions. Most unrecovered loci were filtered out owing to alignment artifacts, and others were missed because of regional deletions in a sister clade that confounded our inference of insertion timing (Figure S4). Newly, we uncovered five additional types of RNA that contribute to chimeric LINE-1 insertions: tRNA, 28S rRNA (or LSU-rRNA), 7SL RNA, Y RNA (HY1 and HY4), and 7SK RNA (Figure 3F). These newly discovered chimera types are less abundant than those previously known, ranging from a few to 21 loci in the genome.

Among these new insertion types, tRNA/LINE-1 chimeras were the most common (n = 21) (Table S3), with molecular evidence that tRNA and LINE-1 were inserted together in a single retrotransposition event, including 9–19 bp TSDs in all eight randomly selected cases and an EN motif at the insertion site for six of them. Interestingly, the length of the tRNA sequences in these chimeric insertions is typically half the length of a full tRNA molecule (Figure 3G), whereas most (53.4%) genomic tRNAs are full-length (Figure S5A). Recent studies have shown that tRNAs are frequently cleaved into smaller fragments, known as tRNA halves (30–40 nt), by specific RNases, such as RNase T2,[57,58] angiogenin (ANG),[59] and RNase L[60] (Figure 3G). Breakpoints of the tRNAs in chimeric LINE-1 insertions closely resemble these cleavage sites, suggesting that they may originate from these tRNA halves. Like 5S rRNA in chimeric LINE-1 insertions, the tRNA sequences are predominantly oriented in the antisense to the LINE-1 counterpart. This suggests that these chimeras may result from twin priming, where two reverse-transcription reactions occur: one at the primary site of 3′ LINE-1 integration and another initiated from the opposite end of a TPRT-induced double-stranded break.[61–64]

The second most commonly occurring new chimeric insertion type is 28S rRNA/LINE-1 sequences (n = 18). 28S rRNA is 5,035 nt long, the longest of the RNA repeats identified, and its chimeric segments are also the largest, with a median size of

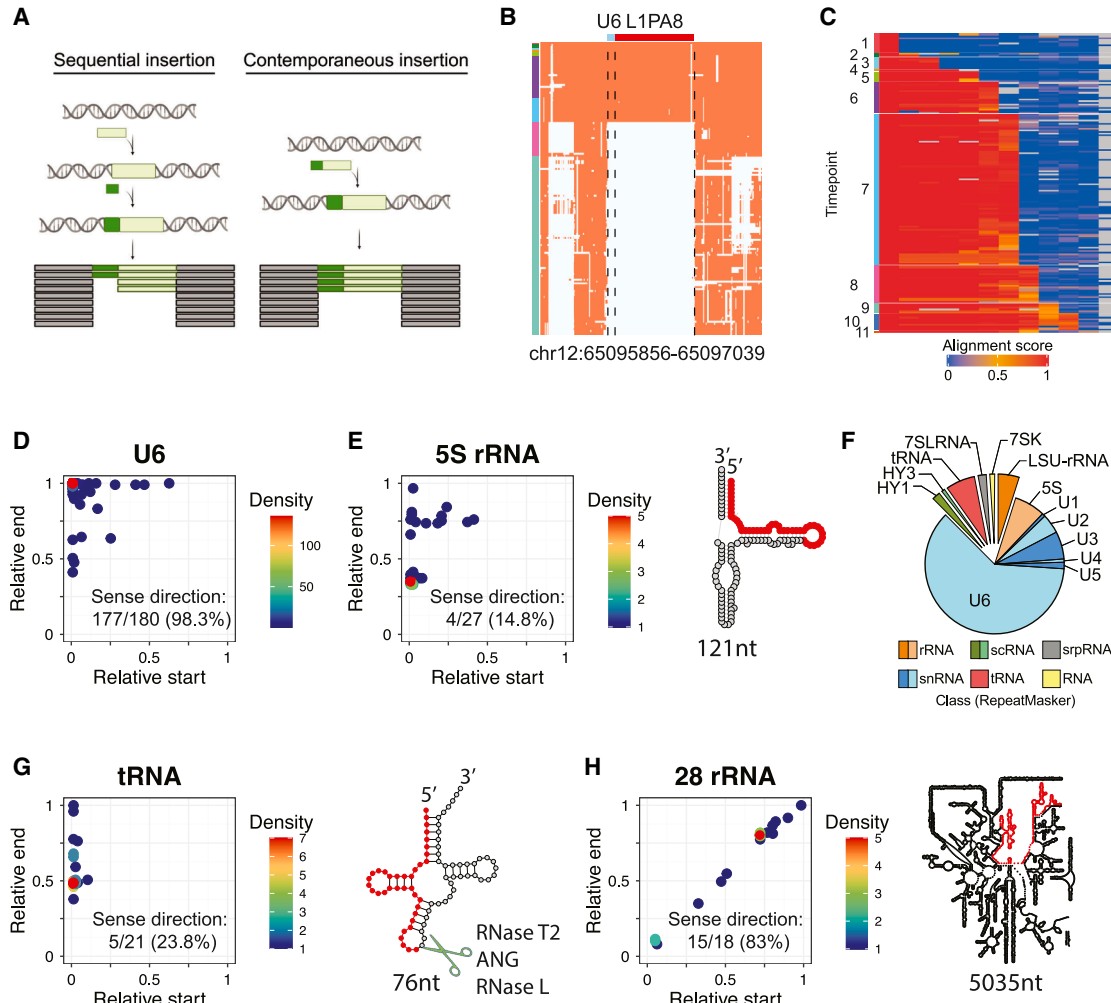

**Figure 3. Characterization of RNA repeat/LINE-1 chimeras**

(A) MSA data reveal the arrival times of two adjacent DNA segments, enabling a distinction between sequential and contemporaneous insertions.

(B) The MSA profile of a chimeric U6/LINE-1 insertion suggests that L1PA8 and U6 were inserted into the genome contemporaneously.

(C) The heatmap shows the human and sister clade/taxon-level alignment score (columns) for all 180 identified chimeric U6/LINE-1 loci (rows), indicating these insertions have spanned a long time frame of LINE-1 activity.

(D) The scatterplot shows the relative start (x axis) and end (y axis) of U6 segments, normalized by the reference U6 length. U6 segments in most chimeric U6/LINE-1 insertions are full-length and in the sense orientation relative to LINE-1.

(E) The 5S segment in chimeric 5S/LINE-1 insertions is often truncated and oriented in the antisense direction relative to LINE-1. In the predicted 5S rRNA secondary structure, the region frequently incorporated into chimeras is highlighted in red (1–45 nt).

(F) The pie chart shows the relative abundance of RNA repeats identified in LINE-1 chimeras. Newly identified types, highlighted in the enlarged section, include 28S rRNA, HY1, HY3, tRNA, 7SK RNA, and 7SL RNA. Others were previously known chimeric LINE-1 insertion types. Types are color coded by their RepeatMasker annotation.

(G) The scatterplot shows the relative start and end sites of tRNA segments in tRNA/LINE-1 chimeras, normalized by the reference tRNA length. Most tRNAs in tRNA/LINE-1 chimeras are half-length and in antisense orientation relative to the LINE-1. In the tRNA secondary structure, the frequently incorporated region (1–37 nt) is highlighted in red.

(H) The scatterplot shows the relative start and end sites of 28S rRNA segments in 28S rRNA/LINE-1 chimeric insertions, normalized by the full-length 28S rRNA. Most 28S rRNA segments are in sense orientation relative to LINE-1. In the 28S rRNA secondary structure, the most frequently incorporated region (3,636–4,115 nt) is highlighted in red.

269 nt (50–482 nt). These 28S rRNA fragments often start around position 3,636 in the consensus 28S rRNA sequence (Figure 3H), a pattern uncommon in other genomic 28S rRNA loci (Figure S5B), although segments from across the 28S sequence were seen incorporated in 28S/LINE-1 chimeras. Small RNAs derived from 28S rRNA have been described, which may be the origin of these 28S sequences.[65] Most of the 28S segments (83%) are oriented in the same direction as their associated LINE-1 elements, suggesting mechanisms of RNA ligation or template switching during TPRT.

In summary, our approach identified numerous new RNA repeats that form chimeric insertions with LINE-1. Many of these have recurrent sequence features predicting molecular mechanisms of recombination with LINE-1.

## *Alu* elements form chimeric insertions with LINE-1

Beyond short RNA species, we identified *Alu* retrotransposons recurrently integrating contemporaneously with and adjacent to LINE-1 3′ ends. To evaluate whether these are chimeric insertions, we analyzed several candidates for sequence hallmarks of LINE-1-mediated retrotransposition (Figure 4A). We found that TSD annotations were difficult to achieve in a subset of older examples owing to nucleotide substitutions (Figures S6A and S6B). To address this, we used consensus sequences derived from MSA to search for flanking TSDs (see STAR Methods; Figure S6C). This method substantially enhanced the recovery of TSDs, evidenced by having more TSDs aligning precisely at the 5′ junction of L1MB elements compared with detection without using consensus sequences (Figure S6D). Benchmarking showed that MSA-based TSD prediction improved with element age (L1PA–L1MB) and refined recognition of ORF2p EN motifs, providing robust molecular evidence for LINE-1 chimeric insertion events (Figures S6E and S6F).

Incorporating this TSD detection method, in total, we identified 452 *Alu*/LINE-1 chimeric insertions all bounded by TSD ($\geq$7 bp) (Figure 4B). Of these, 40% contain EN motifs at their insertion sites (Table S4). Most *Alu* elements were full-length (approximately 300 bp) and oriented in the sense direction relative to LINE-1 (88.5%, Figure 4B). Interestingly, the length distribution of the LINE-1 segments in *Alu*/LINE-1 chimeric insertions closely mirrored the overall length distribution of genomic LINE-1 elements (Figure 4C). The majority (77%) retrotransposed at time point 7, before the divergence of New and Old World monkeys (approximately 60 mya), during a period of *Alu* expansion in primate genomes. The *Alu* families involved in these chimeras are predominantly *Alu*J (61%) and *Alu*S (33%), which were the primary contributors to the *Alu* expansion. Accumulation of *Alu*/LINE-1 chimeras was reduced after time point 5, after the emergence of apes, and became rare after the appearance of *Homo sapiens* (Figure 4D).

To investigate whether *Alu* and LINE-1 components of these insertions were concurrently active when the chimeras formed, we looked for matches in the ages (temporal activities) of their respective subfamilies. This revealed that in 82.8% of chimeric insertions, the associated *Alu* and LINE-1 subfamilies were contemporaneously active and that this matched the period in which the chimera was inserted (Figure 4E). These findings suggest that *Alu*/LINE-1 recombination occurs when both elements are actively retrotransposing and potentially competing for ORF2p interactions. Together, these findings indicate that LINE-1-mediated retrotransposition of *Alu* does not always break *cis* preference[30,66] but can infrequently arise through the formation of *Alu*/LINE-1 chimeric insertions. Additionally, *Alu*/LINE-1 chimeric insertions occurred early in primate evolution but became less frequent in more recent evolutionary history (Figures 4D and S2B).

## 5′ transduction events forming LINE-1 chimeras with mRNAs and lncRNAs

Using our MSA pipeline, we recognized a second class of 5′ transduction events forming LINE-1 chimeras with unique (non-repetitive) RNA sequences. In total, we identified 17 LINE-1 insertions in the hg38 reference assembly containing RNA sequences fused to the 5′ end of LINE-1, including protein-coding gene mRNAs and lncRNAs (Table S5). One example involves a full-length L1PA5 element and 153 bp of the upstream sequence (Figures 5A and S7A). This insertion exhibits all the molecular signatures of LINE-1-mediated TPRT, including a 15 bp TSD flanking both ends of the chimeric sequence, an ORF2p EN motif sequence, and a poly(A) tail (Figure 5A). BLAST analysis revealed that the 5′ non-repetitive sequence maps to the first two exons of a transcript of the mitogen-activated protein kinase kinase kinase 13 (*MAP3K13*) gene in the sense orientation (Figure 5A). This finding suggests a chimeric RNA comprising the first two exons of *MAP3K13* spliced to a LINE-1 with the capacity to encode the requisite proteins for retrotransposition. This chimeric RNA was inserted into the genome at time point 5, concordant with the activity of L1PA5. One model to produce this chimeric RNA would be *cis*-splicing of *MAP3K13* to a downstream intronic LINE-1, although no full-length L1PA5 element is found in the introns of *MAP3K13* in humans or other primates. Such a LINE-1 variant may have existed at some allele frequency in an ancestral population and been subsequently lost. Alternatively, this chimeric insertion could result from a recombination between two distinct RNA molecules. While formally possible, a template-switching event during retrotransposition is not favored given the presence of a splice junction. Another example of a chimeric insertion involves a sequence mapped to an isoform of the fragile histidine triad protein (*FHIT*) gene, an exceptionally large gene spanning 1.5 Mb (Figure 5B). The upstream segment of the chimeric insertion aligns with the first three *FHIT* exons, and the downstream segment is a full-length L1PA4. Like previous cases, we could not identify a potential source LINE-1 in *FHIT* introns.

Across all 17 chimeric insertions, 15 are in the sense orientation relative to the LINE-1 sequence. These concordant orientations are essentially absolute when the 5′ piece maps to the start of the respective gene mRNA (Figure 5C). Moreover, unlike chimeric LINE-1 insertions formed with repetitive RNAs (e.g., U6) or LINE-1 elements in the human genome, which frequently exhibit 5′ truncations, LINE-1 elements in these 5′ transduced events were predominantly close to full-length (Figure 5D). We infer the existence of fusion transcripts encompassing near full-length, protein-coding, and retrotransposition-competent LINE-1 with acquired extra 5′ sequences. Further, we observed that the first exons of some of the associated gene loci contain promoter histone marks and numerous transcription factor binding sites (TFBSs) (Figure S7B), suggesting the possibility that LINE-1 can co-opt novel regulatory elements by this mechanism.

To investigate mechanisms underlying these 5′ transductions, we studied junctions between the appended sequences and their associated LINE-1. Six of the 5′ transduced gene exons terminate precisely at splice donors (Figures 5A, 5B, 5E, 5F, and S8A; Table S5). In four cases, it was not possible to determine whether splice donors were present due to short

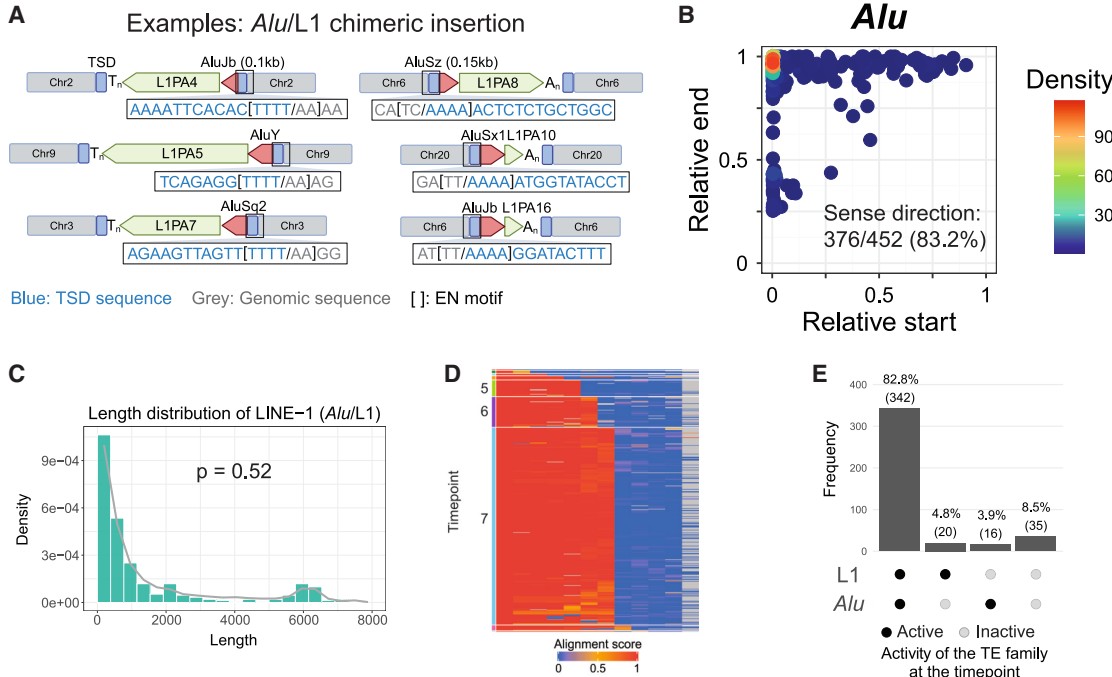

**Figure 4. Chimeric *Alu*/LINE-1 insertions**

(A) Examples of *Alu*/LINE-1 chimeras verified by sequence signatures of TPRT, including 11–17 bp target site duplications (TSDs; highlighted in blue), an ORF2p endonuclease (EN) signature motif (shown in brackets: TT/AAAA for sense orientation or AA/TTTT for antisense orientation), and a poly(A) tail ($A_n$; sense) or poly(T) tail ($T_n$; antisense).

(B) Of 452 *Alu*/LINE-1 chimeric insertions, 83.2% are in the sense orientation relative to LINE-1, with most being full-length insertions.

(C) The length distribution of LINE-1 segments in *Alu*/LINE-1 chimeric insertions (green bars) mirrors L1 length distributions in the genome (gray line); the *p* value is from Wilcoxon's rank sum test.

(D) The heatmap shows the human and sister clade/taxon-level alignment score (columns) for chimeric *Alu*/LINE-1 loci (rows); chimeric *Alu*/LINE-1 insertions largely accrued at time point 7.

(E) For *Alu*/LINE-1 chimeras, co-integrated *Alu* and LINE-1 families were actively retrotransposing at the time of insertion.

untemplated sequences at the junction with LINE-1. In some instances, corresponding splice acceptor sites were identified within the LINE-1 sequence, including a previously known splice acceptor at position 622 of L1PA5[67] (Figures 5E and 5F). However, in some instances, no LINE-1 splice sites were found, particularly when the LINE-1 segment mapped to the first 10 bp of the consensus sequence (Figures 5A, 5B, and S8A) or the splice acceptor site was present within the LINE-1 without the matching donor site (Figure S8B). In aggregate, however, the recurring occurrence of splice sites between the 5′ exons and the downstream LINE-1 indicates *cis*- (same strand) or *trans*-splicing between pre-mRNAs and LINE-1.[68–72]

Potentially relevant to a *trans*-splicing model, the immediate downstream introns of the last exon included in these mRNA/LINE-1 chimeras are often unusually long (Figures 5A, 5B, 5E, 5F, and S7A). Compared with typical lengths of the first and second introns across annotated human genes, these introns associated with chimeric insertions have a median length of 48 kb (163 bp–220 kb) vs. a genome-wide median of 2.2 kb (Figure S8C). These findings suggest a tendency for mRNAs to splice to LINE-1 sequences instead of a distant downstream splice acceptor, forming a LINE-1 chimera that subsequently retrotransposes.

### Co-opted promoters drive expression of 5′ compromised LINE-1

In our evaluation of 5′ transduction events, we identified two nearly full-length (5.5 kb) LINE-1 insertions (Figure 6A), both missing part of the LINE-1 promoter but incorporating upstream sequences mapping to the same gene, Rap1 GTPase-GDP dissociation stimulator 1 (*RAP1GDS1*) (Figure 6B). These two insertions were found on different chromosomes and flanked by non-homologous genomic sequences and distinct TSDs (Figure 6A), indicating independent retrotransposition events rather than an interval duplication. Expanding our search for this specific *RAP1GDS1*-L1PA2 chimeric sequence revealed a total of four insertions in the human reference genome: one insertion on chromosome 10 and three on chromosome 1, owing to segmental duplication of one insertion on chromosome 1.[73]

We identified a putative source L1PA2 for these insertions residing in the second intron of *RAP1GDS1* in a position to permit splicing between a donor in the second *RAP1GDS1* exon and a cryptic splice acceptor 15 bp upstream of the L1PA2. While the L1PA2 has a sequence corresponding to the full length of ORF1 and ORF2, it exhibits an unusual 5′ UTR, with an interstitial deletion corresponding to positions 99–624 in the L1PA2 consensus sequence (Figure 6C). This LINE-1 locus was previously

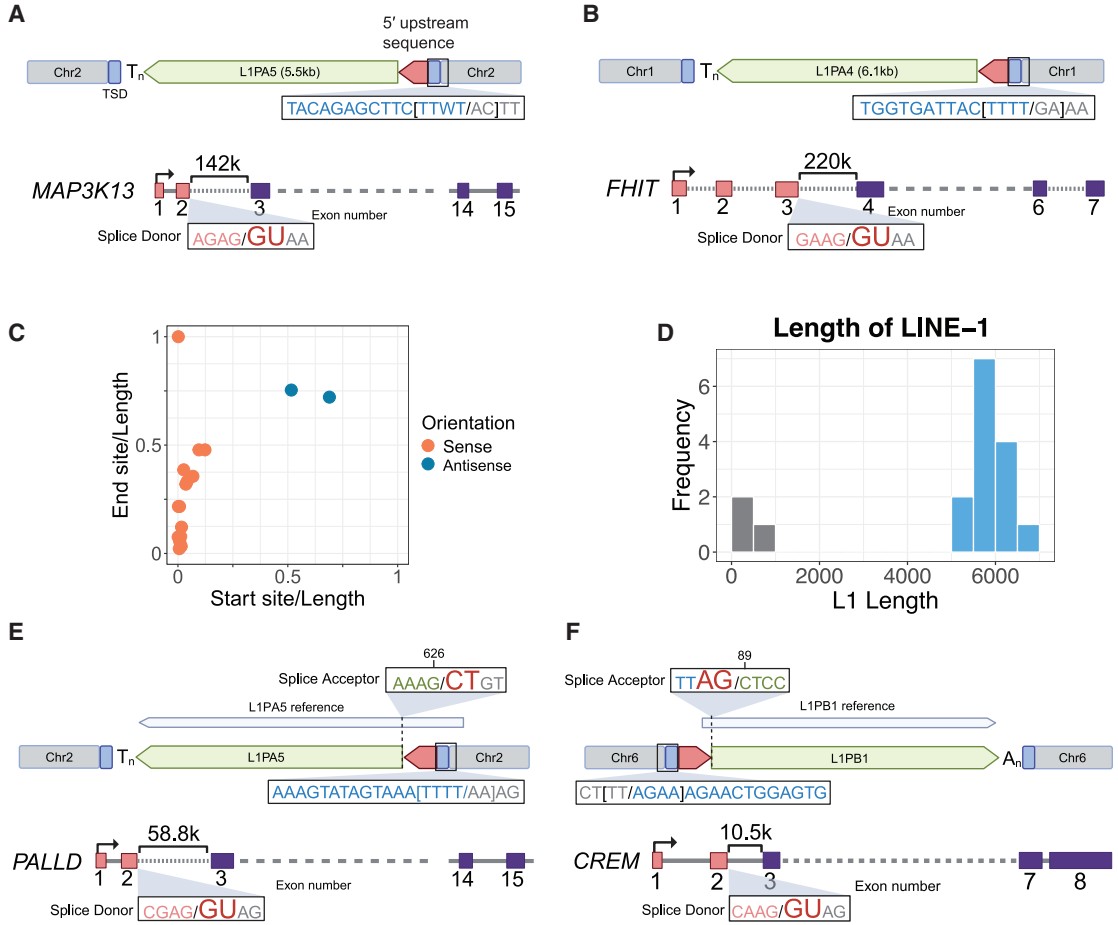

**Figure 5. Chimeric LINE-1 insertions associated with mRNA sequences**

(A) Schematic of chimeric insertion structures, including a 15 bp target site duplication (TSD; highlighted in blue), ORF2p EN motif (in brackets), and poly(T) tail ($T_n$). BLAST analysis aligns the upstream interval with the first two exons of the *MAP3K13* transcript (ENST00000424227), ending at a splice donor site, as diagrammed. The gene spans 15 exons, a subset of which are diagrammed (not to scale). Distances of >40 kb are shown with dashed lines.

(B) Another chimeric insertion includes a 14 bp TSD, an ORF2p EN motif, and a poly(T) tail. The 5' segment corresponds to the first three exons of the *FHIT* transcript (ENST00000488467).

(C) A total of 17 chimeric LINE-1 insertions incorporated segments of protein-coding mRNAs or lncRNAs. Of these, 15 insertions mapped to the beginning of the transcripts, with most in the sense orientation (red) relative to LINE-1.

(D) The majority of chimeric insertions incorporate nearly full-length LINE-1 elements, longer than 5 kb (blue bars).

(E) A chimeric insertion includes a 17 bp TSD, an ORF2p EN-motif, and a poly(T) tail. The upstream sequence maps to the first two exons of the *PALLD* transcript (ENST00000704822), ending at a splice donor site. The LINE-1 segment starts at position 626 of a LINE-1 consensus sequence, adjacent to a splice acceptor at 625.

(F) Another chimeric insertion has all features of ORF2p-mediated retrotransposition (i.e., 16 bp TSD, an ORF2p EN motif, and a poly(A) tail). The upstream sequence maps to the first two exons of the *CREM* transcript (ENST00000354759) and terminates at the exon junction, while the LINE-1 segment maps to position 89 of the L1PB1 sequence, next to a splice acceptor.

identified as a spliced integrated retrotransposed element (SpIRE), and the absence of this region inactivates LINE-1 transcription.[67] Rather than being "dead on arrival," which is typical of SpIREs, it appears that this element co-opted the *RAP1GDS1* gene promoter to restore retrotransposition competence.

Public regulatory element annotations indicate that the *RAP1GDS1* promoter is highly active, supported by extensive transcription factor binding and active histone marks (i.e., H3K4me1, H3K4me3, and H3K27ac) across multiple cell lines (Figure 6D).[74,75] Additionally, *RAP1GDS1* transcription is highly

ubiquitous across tissues (Figure 6E), with particularly prominent expression in the human brain. To explore this further, we analyzed *RAP1GDS1* mRNA expression in brain tissues from humans, chimpanzees, bonobos, and macaques using previously published data[76] and found high expression throughout these non-human primates (Figure S9). We also analyzed publicly available Oxford Nanopore Technology (ONT) long-read transcriptomic sequencing data from 24 human samples, encompassing embryonic and adult tissues as well as induced pluripotent stem cells (iPSCs).[77] We found three RNA reads

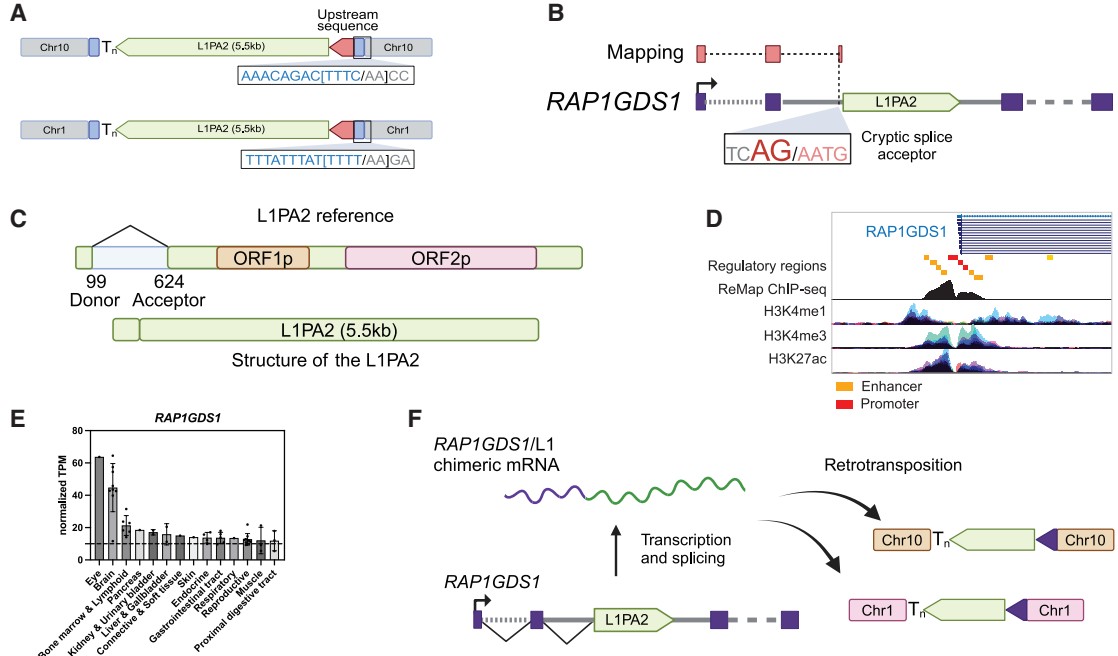

**Figure 6. LINE-1 retrotransposition driven by a co-opted external promoter**

(A) Two chimeric LINE-1 insertions, each containing highly similar LINE-1 and upstream sequences, were identified at two distinct locations. Each insertion is flanked by a distinct TSD (12 or 13 bp) and occurs at an ORF2p EN motif.

(B) The upstream sequences mapped to the first two exons of *RAP1GDS1* (ENST00000508490) and a region directly upstream of a LINE-1 element located within an intron of *RAP1GDS1*.

(C) A more detailed diagram of the source L1PA2 element, which is 5.5 kb in length and contains ORF1 and ORF2. Its 5′ UTR contains an interstitial deletion caused by splicing, inactivating its transcription.

(D) Chromatin immunoprecipitation sequencing (ChIP-seq) data from the ReMap database reveal that the *RAP1GDS1* promoter is bound by numerous transcription factors and has active histone marks: H3K4me1, H3K4me3, and H3K27Ac.

(E) GTEx and Human Protein Atlas data support *RAP1GDS1* expression across multiple tissues, particularly in the brain and eye. Transcripts exceeding 10 transcripts per million (TPM; horizontal line) are generally considered expressed.

(F) A proposed mechanism for the formation of these chimeric LINE-1 insertions. Transcription of the intronic L1PA2 by the active *RAP1GDS1* promoter produces a spliced *RAP1GDS1*/L1PA2 RNA, competent for retrotransposition.

supporting splicing between upstream exons of *RAP1GDS1* and the intronic L1PA2: two from brain tissue and one from testis. This is consistent with the known high expression of *RAP1GDS1* in the brain and offers evidence of the RNA intermediate likely responsible for these retrotransposition events entering the germline.

Together, these findings provide strong evidence that the *RAP1GDS1* promoter drove the expression of an intron-embedded LINE-1 element, compensating for its loss of intrinsic transcriptional activity and rendering it retrotransposition competent (Figure 6F).

## DISCUSSION

### MSA is a powerful tool for TE analysis

Recent genome sequencing efforts across many species[78–80] are making MSA a powerful approach for identifying TE insertions. MSA enables the identification of polymorphic TE insertions within species,[81,82] studies on TE evolution,[83–85] and the discovery of new TEs.[82,86] Additionally, the homoplasy-free nature of TEs and their unambiguous directionality make them

valuable markers for reconstructing phylogenetic relationships between species and among populations.[87–90] Harnessing MSA data, TiMEstamp provides a rapid approach to estimate TE insertion times. By dating these events, we can assess the activity of various TEs across evolutionary timescales. Additionally, TiMEstamp can complement traditional sequence-homology-based annotation methods. For example, non-*Homo sapiens*-specific LINE-1 insertions have sometimes been mislabeled as L1HS, and certain L1M subfamilies are indeed primate specific. By defining insertion timing, TiMEstamp can improve existing TE annotations.

Over millions of years, TEs have shaped our genome, though precisely how they have contributed to speciation and species-specific characteristics is only partially understood. Many primate-specific regulatory regions with cell-type-specific and developmental activities are derived from TEs.[91,92] Timing TE insertions relative to events in speciation could narrow down those involved in primate evolution and guide functional investigations.

Here, we leveraged TiMEstamp and MSA to identify chimeric LINE-1 insertions without relying on specific knowledge of those fusions *a priori*. MSA enabled us to efficiently identify adjacent

**A**

RNA processing

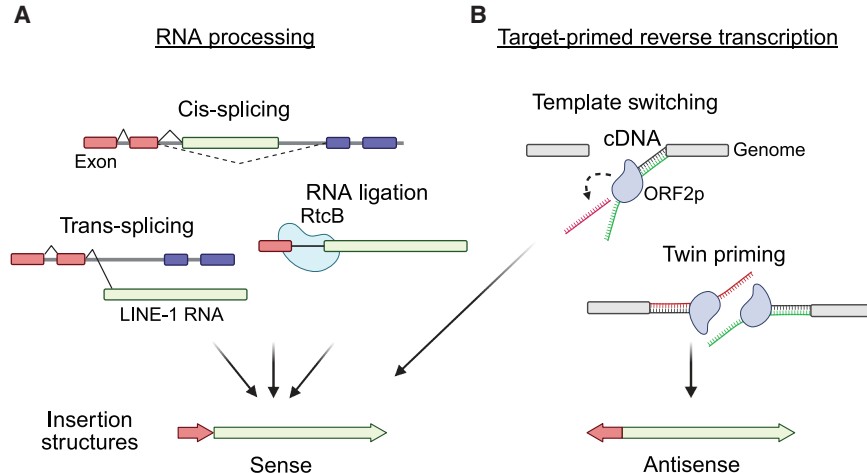

**B**

Target-primed reverse transcription

**Figure 7. Known and proposed mechanisms for the formation of LINE-1 chimeric insertions**

(A) RNA processing produces fusion transcripts that are subsequently retrotransposed. *Cis*-splicing: an intronic LINE-1 is misspliced into a surrounding gene; *trans*-splicing: a LINE-1 RNA is spliced with another transcript; and RNA ligation, via RtcB RNA ligase.

(B) Target-primed reverse transcription (TPRT) forms the chimeric integration. Template switching: the ORF2p reverse transcriptase jumps from its primary LINE-1 RNA template to another RNA. Twin priming: two reverse-transcription reactions are initiated on opposite ends of a double-strand DNA break and joined in the resolution of the insertion. Chimeras formed by RNA processing or RT template switching generate insertions with non-LINE-1 and LINE-1 sequences in the same orientation; twin priming results in chimeric segments in opposite orientations.

insertions occurring contemporaneously and improved recovery of the ORF2p EN cut-site motif and TSD flanking insertions, allowing us to conclude, in many instances, that the sequences were inserted in a single retrotransposition event. Improvements in MSA alignment tools[93] and the development of TE detection methods from MSA[82] will no doubt continue to accelerate this field.

## Identification of new RNA repeats forming chimeric insertions with LINE-1

We found an additional five types of repetitive RNAs that can form chimeric insertions with LINE-1, including Y RNA, tRNA, 7SL RNA, 7SK RNA, and 28S rRNA. Similar to previously known small, repetitive RNA that form LINE-1 chimeric insertions,[33,94,95] these newly identified RNA repeats are known to integrate into the genome as pseudogenes through LINE-1-mediated TPRT without forming a chimera.[6,32,96–100] Interestingly, the first reported 28 rRNA pseudogene locus in human was identified right next to a LINE-1 element,[101] and our analyses now show that this is in fact a chimeric insertion (Table S3). Notably, previous studies have shown that U1-U6 RNA, 7SK RNA, Y RNA, and 7SL RNA interact with ORF1p,[99] and their association with L1 RNPs may potentiate engagement with the ORF2p RT. Some of these RNA repeats have been incorporated into SINEs in various organisms, which can propagate using LINE machinery, for example, tRNA,[102–104] 7SL RNA,[105,106] 5S rRNA,[107–109] 28S rRNA,[110] and U1/2 snRNA.[111] Primate-specific *Alu* SINEs originated from 7SL RNA after the primate/Glires split[105,112,113]; rodent-specific B1 SINEs arose from a tRNA sequence.[114]

Structural features of chimeric RNA repeat/LINE-1 insertions provide clues about mechanisms underlying their formation. The sense orientation bias relative to LINE-1 of U6, U3, and 28 rRNAs suggests the formation of these chimeras through RNA ligation or a template switch during the primary TPRT; in contrast, antisense-orientation bias may indicate twin priming as a mechanism (Figure 7).

Recombination between various RNAs and existing retroelements can form novel composite mobile elements. For instance, the hominid-specific SVA element is formed by the fusion of an LTR sequence, a VNTR, and an *Alu*-like sequence.[50] Recently, it was discovered that retrotransposable element (RTE)-type LINE, a subfamily with a short 5′ UTR and a single ORF,[115] can form chimeras with tRNA or 5S rRNA to generate new SINEs.[116] This illustrates that recombination between RNA species and retroelements, creating novel chimeric sequences, can drive genome evolution by contributing new families of mobile genetic elements.

## Chimeric insertions with partial non-genetic RNAs

Our study identified a diverse collection of LINE-1 chimeric insertions involving so-called non-genic RNAs. Many of these RNAs, including tRNA, 5S rRNA, 28S rRNA, and U3 snoRNA, have been recurrently incorporated as fragments fused to the LINE-1 sequence. For tRNAs, we often see tRNA halves joined to LINE-1; it is known that cells cleave tRNAs into these half-length forms,[117,118] suggesting they might be truncated before reverse transcription at the insertion site. tRNAs are also most commonly oriented oppositely of the associated LINE-1, supporting a distinct reverse-transcription reaction reminiscent of the twin-priming model proposed for ORF2p (Figure 7B).[61] Interestingly, previous studies have demonstrated that tRNA fragments can self-prime ORF2p-mediated retrotransposition[96] and serve as primers for other retrotransposons in various organisms.[119–123] Additionally, their high abundance in cells[124] may enable them to exploit exposed DNA ends and reverse transcribe. Although less is known about the reverse transcription of 5S rRNA, U3 snoRNA, or 28S rRNA fragments, we see these incorporated in LINE-1 chimeric insertions as well.

## *Alu* forms chimeric insertions with LINE-1

Our study revealed that *Alu* elements can form chimeric insertions with LINE-1. In these *Alu*/LINE-1 chimeras, the *Alu* portion is predominantly sense oriented and typically full-length (~300 bp), which matches their general length distribution in the human genome. LINE-1 portions in these chimeric insertions

tend to be 5′ truncated, as they are elsewhere. Like U6/LINE-1 chimeras, these features may result from RNA ligation before retrotransposition (Figure 7A) or template switching during TPRT (Figure 7B). The observation that LINE-1 most frequently forms *Alu*/LINE-1 chimeric insertions with concurrently active *Alu* subfamilies highlights a potential requirement for *Alu* expression or other functionality. It is unclear if the formation of these chimeras depends on ribosomal stalling by *Alu*, which normally permits it to co-opt ORF2p.[66]

### LINE-1 may form chimeric insertions with gene mRNAs through *trans*-splicing

Using TiMEstamp, we identified 17 mRNAs forming chimeric insertions with LINE-1. Notably, 15 of the respective gene loci do not contain intronic LINE-1 sequences, suggesting that RNA recombination in *trans* may precede reverse transcription and genomic integration. Supporting a *trans*-splicing model (Figure 7A), most of these chimeras incorporate the first exon of a transcript in sense orientation with the LINE-1, and splice donor and acceptor sites are found at many exon-LINE-1 junctions. The events appear to preferentially involve gene exons immediately upstream of long introns, consistent with prior findings that longer introns are prone to *trans*-splicing.[125] These chimeric insertions often incorporate full-length or nearly full-length LINE-1 elements, retaining the protein-coding capacity for ORF1p and ORF2p. Thus, these events juxtapose the first exon of a gene, sequences often enriched with TFBSs, with LINE-1 protein-coding sequences, creating possibilities for the evolution of the LINE-1 promoter and evasion of retroelement-targeted silencing mechanisms.

In contrast, the mRNA/LINE-1 chimeras with antisense-oriented mRNA did not incorporate a first exon or have features suggestive of splicing and may be more consistent with twin priming.

### Promoter co-option by an intronic LINE-1

Two of the 17 mRNA/LINE-1 chimeras share the same sequence, although they arose through independent TPRT events. They likely derived from a *cis*-splicing event between 5′ exons of *RAP1GDS1* and an intronic LINE-1 element in that gene. This L1PA2 was previously reported to contain an internal splice that deletes approximately 500 bp of its 5′ internal promoter[67] and eliminates multiple TFBSs,[126–130] effectively abolishing its transcription. In this case, however, the element restored retrotransposition and transcriptional activity via the *RAP1GDS1* promoter. This observation builds on prior recognition that intronic LINE-1 elements can be transcribed by external promoters[6,126,131–135] and incorporate 5′ transductions into new genomic loci. To our knowledge, though, this is the first instance of a transcriptionally disabled, intronic LINE-1 co-opting an external promoter and splicing mechanisms to facilitate retrotransposition.

Co-opting an external promoter and 5′ splicing of the *RAP1GDS1* locus (Figure 7A) may enable this element to evade LINE-1-specific silencing. For example, the human silencing hub (HUSH) complex, which targets promoters of intronless LINE-1,[136,137] may not recognize intron-containing *RAP1GDS1*/L1PA2 transcripts. Krüppel associated box

(KRAB)-containing zinc-finger proteins[138,139] and DNA methylation, which typically silence the LINE-1 promoter,[132,134] would not affect an external promoter. A recent study suggested that during spermatogenesis, reduced expression of the RNA-binding protein SAFB, which normally suppresses the inclusion of intronic LINE-1 sequences during splicing, may create a permissive environment for intronic LINE-1 incorporation and chimeric transcript formation.[140] Such a window, together with the active and ubiquitous transcription of *RAP1GDS1*, may have potentiated retrotransposition of this element in the germline and contributed to somatic mosaicism in ancestral primates until internal mutations affected the LINE-1 ORFs.

### Chimeric insertions may have driven LINE-1 evolution

Transcription driven by the LINE-1 internal promoter is the first critical step in the LINE-1 life cycle. Successively active LINE-1 subfamilies often have distinctive 5′ UTRs compared to preceding subfamilies with declining activity, whereas the ORF2 and ORF1 sequences remain relatively conserved among primate LINE-1 families. The acquisition of novel 5′ UTRs has slowed only recently, following the emergence of L1PA8.[5] The punctuated evolution of the 5′ UTR without transitional intermediate sequences suggests that LINE-1 may employ special mechanisms to drive this diversification. We speculate that these dramatic "swaps" of its 5′ UTR by recombining with other RNAs to form chimeric insertions in the genome could introduce entirely new TFBSs for *trans*-activators or permit LINE-1 to bypass host cell suppressors and explain its evolutionary success. We expect that studies of chimeric and composite retroelements will continue to yield insights into TE adaptability and evolution.

### Limitations of the study

TiMEstamp relies on MSAs of genome assemblies to estimate the age of mobile element insertions, and its accuracy is therefore contingent on the quality of the MSA. Because MSAs can contain ambiguous or pseudo-alignments, particularly across evolutionarily distant species, some timing estimates may be inaccurate. Additionally, potential mechanisms for the formation of chimeric LINE-1 insertions were inferred from their structural features and prior knowledge. Experimental studies will be needed to test these predictions.

### RESOURCE AVAILABILITY

#### Lead contact
Further information requests should be directed to the lead contact, Kathleen H. Burns (kathleenh_burns@dfci.harvard.edu).

#### Materials availability
No unique reagents are generated in this study.

#### Data and code availability
The MSA in this study can be accessed from UCSC Genome Browser (https://hgdownload.soe.ucsc.edu/goldenPath/hg38/multiz470way/). All processed data are available from the lead contact. The TiMEstamp R package is available at Zenodo (https://doi.org/10.5281/zenodo.17466832) and GitHub (https://github.com/ctl43/TiMEstamp).

## Article

**CellPress**

## ACKNOWLEDGMENTS

K.H.B. and her lab are supported by NIH grants (R01CA240816, R01CA276112, R01CA289390, and UG3NS132127). C.-T.L. is supported by an American Cancer Society Postdoctoral Fellowship Award (PF-23-1149403) and NIH Pathway to Independence Award (K99) (5K99GM157510). Some figures were created using BioRender.com.

## AUTHOR CONTRIBUTIONS

C.-T.L. and K.H.B. conceived the project and wrote the manuscript. C.-T.L. performed computational analyses.

## DECLARATION OF INTERESTS

The authors declare no competing interests.

## DECLARATION OF GENERATIVE AI AND AI-ASSISTED TECHNOLOGIES IN THE WRITING PROCESS

In this work, C.-T.L. used ChatGPT (OpenAI) to improve code readability, efficiency, and documentation. All outputs were carefully edited and validated by C.-T.L., who takes full responsibility for the content of this publication.

## STAR★METHODS

Detailed methods are provided in the online version of this paper and include the following:

- KEY RESOURCES TABLE
- METHOD DETAILS
  - Preparing repeat datasets
  - Preparing multiple sequence alignments
  - Building a census of known chimeric LINE-1 insertions
  - TiMEstamp: Calling presence/absence of LINE-1 from MSA data and predicting their insertion times
  - Defining insertions 5′ of LINE-1 and predicting their insertion times
  - Predicting chimeric LINE-1 insertions
  - Annotation of the unannotated sequence of the chimeric LINE-1 insertions
  - Determining target site duplications (TSDs)
  - ORF2p endonuclease (EN) motif analysis
  - Transposable element age analysis
  - Transcriptomic data analysis
  - RNA secondary structure prediction
- QUANTIFICATION AND STATISTICAL ANALYSIS

## SUPPLEMENTAL INFORMATION

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

## STAR★METHODS

### KEY RESOURCES TABLE

| REAGENT or RESOURCE | SOURCE | IDENTIFIER |
|---|---|---|
| **Deposited data** | | |
| Repeat annotations for hg38 (Repeat Library 20140131) | RepeatMasker[44] | https://www.repeatmasker.org/genomes/hg38/rmsk4.0.5_rb20140131/hg38.fa.out.gz |
| Multiple sequence alignment across 470 mammalian genome assemblies | UCSC genome browser | https://hgdownload.soe.ucsc.edu/goldenPath/hg38/multiz470way/ |
| Gencode (version 46) | Frankish et al. 2023[55] | https://www.gencodegenes.org/human/release_46.html |
| RNAcentral (version 24) | The RNAcentral Consortium 2019[56] | https://ftp.ebi.ac.uk/pub/databases/RNAcentral/releases/24.0/ |
| Rfam | The RNAcentral Consortium 2019[56] and Ontiveros-Palacios et al. 2024[141] | https://ftp.ebi.ac.uk/pub/databases/RNAcentral/releases/24.0/rfam/ |
| Primate transcriptomic data | Khrameeva et al. 2020[77] | GEO: GSE127898 |
| Long-read sequencing data of human tissues | Kaur et al. 2024[78] | BioProject: PRJEB81685 |
| **Software and algorithms** | | |
| TiMEstamp | This paper | https://github.com/ctl43/timestamp Or Zenodo: https://doi.org/10.5281/zenodo.17466832 |
| Phylogenetic Analysis with Space/Time (PHAST) | Hubisz et al. 2011[142] | http://compgen.cshl.edu/phast/ |
| BLAST (version 2.12.0) | Altschul et al. 1990[143] | https://ftp.ncbi.nlm.nih.gov/blast/executables/blast+/2.12.0/ |
| Bedtools (version 2.30.0) | Quinlan 2014[144] | https://github.com/arq5x/bedtools2/releases/tag/v2.30.0 |
| R2DT | McCann 2021[145] | https://rnacentral.org/r2dt |

## METHOD DETAILS

### Preparing repeat datasets

Repeat annotations for the hg38 genome assembly (version: Repeat Library 20140131) were obtained from www.RepeatMasker.org and served as the basis for recognizing LINE-1 sequences and other interspersed repeats in our analysis pipeline, as illustrated in Figure 1. Over evolutionary time, LINE-1 sequences can be fragmented by other transposable element insertions (i.e., 'nesting'). To address this, distinct LINE-1 fragments putatively derived from the same insertion were grouped based on their repeat ID predicted by RepeatMasker.[44] The orientation of each LINE-1 segment was determined with respect to the fragment closest to the 3′ end of the LINE-1 consensus sequence. To predict candidate chimeric LINE-1 insertions, we excluded LINE-1 elements with 5′ inversions by requiring the same orientation across all fragments. The 5′ most end of each LINE-1 insertion was then used for chimeric LINE-1 predictions. For evolutionary studies in Figure 2, LINE-1 fragments with the same repeat ID were similarly consolidated into potentially single insertions, although orientation was not considered. Finally, where needed, consensus sequences representing each LINE-1 family were retrieved from Khan et al. (2006).[5]

### Preparing multiple sequence alignments

Multiple sequence alignment (MSA) data in multiple alignment format (MAF), encompassing 470 mammalian genome assemblies from 444 species, were prepared by Hiller and colleagues as described in Hecker et al.[39] and retrieved from the UCSC Genome Browser (https://hgdownload.soe.ucsc.edu/goldenPath/hg38/multiz470way/), together with the phylogenetic tree.[146] MAF files were split by species, and gaps relative to the human reference genome assembly (hg38) were removed using the Phylogenetic Analysis with Space/Time (PHAST) program commands mafSpeciesSubset -keepFirst and msa_view –gap-strip 1.[147] Since precise alignment within the MSA was not critical for our analysis, we focused on unaligned regions to reduce data size. These unaligned regions, indicated by asterisks (*) or hyphens (−) in the resulting FASTA files, were extracted and annotated in BED file format. Additionally, we excluded non-ape sister clades containing fewer than 10 species, as smaller clades are less effective at reducing noise

through averaging, especially when evolutionarily distant from humans. After filtering, 464 genome assemblies from 438 species were retained.

### Building a census of known chimeric LINE-1 insertions

To compile a census of chimeric LINE-1 insertions with known structures (e.g., U1-6 small nuclear RNA fused to 5′ LINE-1 and 5S ribosomal RNA fused to 5′ LINE-1), we identified LINE-1 where these short RNA species were annotated by RepeatMasker within 50bp upstream of the LINE-1 element. MSA profiles of these loci were manually inspected to confirm contemporaneous insertion of the small RNA and LINE-1.

### TiMEstamp: Calling presence/absence of LINE-1 from MSA data and predicting their insertion times

Presence or absence of a LINE-1 insertion across different mammals allows us to place its arrival time with respect to speciation. Before classifying a LINE-1 as "present" or "absent" in each species, we calculated the aligned portion to annotated LINE-1 in the human genome assembly, regardless of mismatches and averaged the aligned portion across species within the sister clade. For each locus, we ensured that "absent" calls would be restricted to genomes with pre-insertion sequence; and those harboring a larger genomic deletion or lack of the region encompassing the LINE-1 (e.g., missing >1000bp of flanking genomic DNA) were classified as "missing" the LINE-1 insertion site and excluded from the "present" vs. "absent" dichotomy. All species in a non-ape sister clade were marked as missing an insertion site if the site was missing from 90% of clade members. For evolutionary analyses in Figure 2, we labeled an insertion as "present" in a sister clade if the average aligned portion exceeds 50% of annotated LINE-1. For chimeric LINE-1 analysis, we used a threshold of 65% alignment (two-thirds) to classify the LINE-1 adjacent segment as "present", while genomes with no or shorter homologous intervals were classified as "absent" for the segment. These thresholds were empirically chosen by examining their ability to recover known chimeric LINE-1 insertions, using a manually curated benchmark set of 253 cases (U1–6 and 5S) whose MSA profiles were individually inspected. Presence of a LINE-1 sequence in sister clades was used to infer the earliest common ancestor with the insertion and thus the timing of its introduction into mammalian genomes. The TiMEstamp R package is available at: https://github.com/ctl43/TiMEstamp and Zenodo (https://doi.org/10.5281/zenodo.17466832).

### Defining insertions 5′ of LINE-1 and predicting their insertion times

Sequence insertions 5′ of LINE-1 were found by identifying 'gaps' between LINE-1 insertions annotated in the human genome assembly and upstream sequences present in genome assemblies lacking the LINE-1 insertion. Gaps shorter than 1000bp and longer than 25bp were included and were aggregated together into gap clusters if they were within 20bp of each other. For each gap cluster present in at least 15 genome assemblies, the gap supported by the most genomes (with a minimum of 5 species) was selected as the representative gap. The shortest segment was designated as the representative upstream region of LINE-1. The timing of the introduction of this sequence was inferred using the same method applied for LINE-1, with a homology threshold of 65%. Regions with an average alignment below 65% were classified as absent.

### Predicting chimeric LINE-1 insertions

To identify potential chimeric LINE-1 insertions, we compared the predicted arrival time of the LINE-1 element and its 5′ adjacent insertion. Where the two appear contemporaneous (i.e., are inferred to arrive within the same time interval), the pair was classified as a candidate chimeric LINE-1 insertion. These candidates were filtered for alignment noise by excluding instances where more than 10% of genome assemblies in the non-ape sister clades classified as "absent" for the LINE-1 element, exhibited ≥25% alignment for LINE-1 sequence. MSAs for all presented data were manually inspected, using either the 470-way alignments generated by MULTIZ from the Hiller lab or the 447-way mammalian MSA produced by Cactus from UCSC Genome Institute and the Farh lab or both.[39,80,146]

### Annotation of the unannotated sequence of the chimeric LINE-1 insertions

To discern the origins of sequences 5′ adjacent to chimeric LINE-1, we ran BLAST[141] searches of the upstream sequences against RNAcentral,[56] Gencode (version 46),[55] and Rfam databases.[142] Alignments exceeding covering >75% of the query sequence and exceeding 75bp were prioritized. The best alignment was then determined by considering (in order): % of the query sequence covered by the alignment, the length of the alignment, and the pairwise alignment identity. For chimeric insertions fusing protein-coding mRNAs or long non-coding RNAs (lncRNAs) with LINE-1, annotations were manually verified.

### Determining target site duplications (TSDs)

Flanking TSDs offer molecular evidence that compound or complex insertions occurred together in a single target primed reverse transcription (TPRT) event. To delineate TSDs at insertion junctions while permitting for some single nucleotide substitutions since the insertion, sequences flanking LINE-1 insertions were extracted from MSAs and converted into International Union of Pure and Applied Chemistry (IUPAC) ambiguity codes. For TSD prediction, sequences extracted by bedtools[143] encompassed 75bp upstream and 25bp downstream of the 5′ junction, and 25bp upstream and 75bp downstream of the 3′ junction. The IUPAC-converted 5′ and 3′ sequences were aligned using a Smith–Waterman local alignment algorithm[144] to detect regions of similarity at the junctions. A modified scoring matrix, adapted from the DNAfull scoring matrix,[145] was employed. In this scoring matrix, we reduced the match score

from 5 to 3, and increased the mismatch penalty from −4 to −10, effectively suppressing the extension of alignments beyond mismatched regions. Additionally, the gap opening penalty was set to −50 to minimize the occurrence of gaps in the alignments. For LINE-1, predicted TSDs were considered high-quality if their lengths ranged from 10 to 30bp and the end of the TSD at the 5′ junction was located within 3bp upstream or downstream of the junction of the LINE-1 element. For chimeric LINE-1 insertions, the TSD length requirement was adjusted to 7 to 20 bp, and the end of the TSD at the 5′ junction was allowed to occur within a broader range, between 40bp upstream and 25bp downstream of the predicted 5′ junction. In cases where the 3′ end of the LINE-1 was poorly annotated due to polyA homopolymers in the preinsertion sequence, TSDs were manually curated.

### ORF2p endonuclease (EN) motif analysis

For the ORF2p EN motif analysis of the L1MB4 family, we extracted sequences spanning the 5′ junction of the TSD at the 5′ end of LINE-1, including 4bp upstream and downstream of the junction. These sequences were then analyzed for motif enrichment using the MEME Suite.[148] For chimeric insertions, 2 bps upstream of the TSD at the 5′ end of LINE-1 were identified as potential starts of the ORF2p EN motif.

### Transposable element age analysis

Transposable element (TE) age was calculated using the percentage divergence values from a subfamily consensus sequence provided by RepeatMasker. Using the Jukes-Cantor model,[149] the divergence was corrected for multiple substitutions, and the age was estimated in millions of years (Myr) based on a substitution rate of 0.17% per million years.[5]

### Transcriptomic data analysis

Oxford Nanopore Technology (ONT) long-read sequencing data were downloaded from the NIH BioProject (Accession number: PRJEB81685).[77] Sequencing reads were aligned to the *RAP1GDS1* segment of the *RAP1GDS1*/LINE-1 chimeric insertion using minimap2.[150] Extracted sequences were then aligned to the L1PA2 sequence to identify chimeric transcripts. The expression of *RAP1GDS1* in primates was retrieved from a precomputed gene expression matrix in the Gene Expression Omnibus (GEO) (Accession number: GSE127898).[76]

### RNA secondary structure prediction

5S rRNA, tRNA and 28S rRNA (LSU-rRNA) sequences were extracted from Dfam[151] and analyzed using RNA Secondary Structure Determination Tool (R2DT),[152] a tool integrated with RNAcentral, to predict and visualize RNA secondary structures.

### QUANTIFICATION AND STATISTICAL ANALYSIS

Details on statistical tests and quantification methods are provided in the figure legends and in the method details section.

