## [Document S2. Transparent peer review records for Law et al. · Cell Genomics]

Summary

Initial submission: Received: 4/18/25

Scientific editor: Laura Zahn

First round of review: Number of reviewers: 2
Revision invited: 5/30/25
Revision received: 9/1/25

Second round of review: Number of reviewers: N/A
Accepted: 1/21/26

Data freely available: Yes

Code freely available: Yes

This transparent peer review record is not systematically proofread, type-set, or edited. Special characters, formatting, and equations may fail to render properly. Standard procedural text within the editor's letters has been deleted for the sake of brevity, but all official correspondence specific to the manuscript has been preserved.

Referees' reports, first round of review

Reviewer #1: Note: This manuscript was co-reviewed by the PI and a graduate student with an interest in this area of research. All comments by the graduate student were reviewed by the PI and consolidated into a single review to eliminate the same comments being made twice in several cases.

This is a well-written paper offering substantial insight into the mechanisms of L1 insertion in humans and other mammals. We see only minor needs for revision.

Combined comments:

Introduction, p3 - Change compliment to complement.

Introduction, p5 - Pet peeve of mine. Change "(MSA) of 464 genomes" to "(MSA) of 464 genome assemblies". Genomes are real things that exist in cells. You didn't align those. You aligned our best approximations of those genomes, the genome assemblies. Same issue on p10, should be "where more than 10% of assemblies in the non-ape...". Other instances were also observed and should be corrected. In each case ask yourself if you're writing about the actual genome that exists in the real world or if you're writing about the genome assembly that you manipulated for your analyses.

This doesn't bother me but it may be against rules for the publication. Figure 2 is referenced twice before figure 1.

p10 - Change to "these upstream sequences were subjected to BLAST..."

p16 - "Meanwhile, the LINE-1 5' truncation in these chimeras is more pronounced than considering the length distribution of all genomic LINE-1" Is 'than' supposed to be 'when'?

Methods:

TiMEstamp: Calling presence/absence of LINE-1 from MSA data and predicting their insertion times.
Page 7: "we used a threshold of 65% alignment (two-thirds) to classify a segment as "present", while no or shorter homologous intervals were classified as "absent".
Why 65%? Also, I think that in this whole section will be interesting to justify parameters like the 1000 bp gap threshold or other thresholds with empirical or simulation-based evidence.

Results:

Page 14: There is a typo in the title Results, change to Results.

Page 17: "we are better positioned to see recurrent patterns for less common LINE-1 chimeras..." Change to patterns.

Discussion:

Page 27: "Consistent with previous studies, we found U6 segments in U6/LINE-1 chimeric insertions are predominantly in full-length and oriented in the sense direction with respect to the associated LINE-1."
Remove "in" before "full-length."

MSA is a powerful tool for TE analysis:

This section could be improved by adding the evolutionary implications of the tool.

Identification of new RNA repeats forming chimeric insertion with LINE-1

This section will benefit from having a short explanation or idea to explain why certain RNAs show specific patterns.

Chimeric insertions may have driven LINE-1 evolution:

This section should elaborate on the mechanistic basis for their speculation that chimeric insertions drive LINE-1 evolution.

Figure 2: Panel D's heatmap is difficult to interpret; clarifying the phylogenetic relationship would be beneficial.

Figure 3: Panel F's pie chart is hard to interpret due to small text and similar colors, and Increase font sizes in legends in scatter plots.

Figure 5: Figure 5 has a lot of valuable data, but its density may make it hard for readers to understand. I suggest splitting it into two figures to make it

Reviewer #2: This is an excellent manuscript featuring significant original research. Here are some suggestions:

1. The manuscript is quite fundamental and, as such, is somewhat distant from practical applications. There are several RNA-based studies on eQTL and regulatory findings of TEs that the authors might explore and discuss in relation to their impact. This would enhance the translation of the study.

2. Here is one manuscript (PMID: 37511314) illustrating the inclusion and severity of SVAs on a genome-wide scale. This paper serves as a good example to demonstrate the clinical significance of this topic.

Authors' response to the first round of review

Response to Reviewers

We are sincerely grateful to the reviewers for their constructive and thoughtful comments, which have improved our manuscript. We have carefully revised the text, figures, and methods in response to the suggestions, and we believe these changes have strengthened both the clarity and impact of our work. Below, we provide a detailed, point-by-point response, with all corresponding revisions highlighted in the manuscript using track changes.

Reviewer #1

General comment:

We are grateful to Reviewer #1 for the positive evaluation of our work and for the detailed, constructive suggestions. We have addressed all minor revisions and made clarifications where necessary.

Introduction

- "Change compliment to complement"

Corrected as suggested.

- "Change 'MSA of 464 genomes' to 'MSA of 464 genome assemblies' (and similar cases)"

We have revised throughout the manuscript to accurately refer to "genome assemblies."

- "Figure 2 is referenced twice before Figure 1"

Corrected the figure order and references.

- "p10 – Change to 'these upstream sequences were subjected to BLAST...'"

Corrected as suggested.

- "p16 – 'Meanwhile, the LINE-1 5' truncation in these chimeras is more pronounced than considering the length distribution of all genomic LINE-1' – Is 'than' supposed to be 'when'?"

Corrected to "when."

Methods

- "Why 65%? Justify parameters such as 65% alignment [coverage] threshold, 1000 bp gap [missing alignment] threshold, etc."

We thank the reviewer for raising this point. To systematically evaluate thresholds for defining the presence of a sequence and for determining whether a DNA segment is missing, we performed a grid search across a range of values for both parameters (see Figure for Reviewers, below). The performance of each threshold combination was assessed using a manually curated set of 253 known chimeric LINE-1 insertions (U1–6 and 5S) as a benchmark, for which MSA profiles were individually inspected. For each threshold setting, we measured the recovery rate of the curated set (A) and, in parallel, Response to Reviewers calculated a selectivity index (B), defined as the inverse of the proportion of loci passing the filter, such that higher values indicate greater stringency. Finally, we used the ratio of recovery rate to the passing percentage (recovery-to-passing) as an indicator of overall performance (Figure C), ensuring that sensitivity for true positives was maintained while effectively excluding non-target loci.

Recovery rates remained consistently high (90–95.2%) across coverage thresholds of 25–65%, with the highest recovery observed at 50% coverage. In contrast, selectivity declined gradually with increasing segment thresholds (row) at each coverage cutoff, with many false positive loci identified at coverage thresholds smaller than 60% (column) and greater selectivity observed at a missing segment threshold of 500 bp combined with coverage thresholds of 75–95%. Evaluation of the recovery-to-passing ratio indicated optimal performance with segment thresholds of 500 bp and coverage

thresholds of 40–85%, as well as with segment thresholds of 1000 bp and coverage thresholds of 50–65%. Because upstream segments of potential chimeric insertions can exceed the 500 bp cutoff while the known chimeras used for benchmarking have uniformly shorter 5' segments, we focused subsequent analyses using the 1000 bp threshold. Within this range, we selected 65% coverage as the threshold for detecting chimeric LINE-1 insertions, prioritizing specificity. For the evolutionary analysis of transposable elements, we applied a less stringent coverage cutoff of 50% (Figure D).

Figure for Reviewers: Evaluation of threshold combinations. (A) Heatmap showing recovery of known chimeric insertions across a range of threshold combinations. (B) Heatmap showing selectivity, defined as the inverse percentage of passing LINE-1 loci. (C) Heatmap showing the recovery-to-passing ratio, used as a performance indicator to preserve sensitivity for true positives while excluding non-target loci. (D) With a segment threshold of 1000 bp, the recovery-to-passing ratio was maximized at coverage thresholds of 50–65%.

Results

- “Typo: Results → Results”

Corrected.

- “p17 – ‘pattens’ → ‘patterns’”

Corrected.

Discussion

- “Remove ‘in’ before ‘full-length’ (p27)”

Corrected.

- “MSA is a powerful tool for TE analysis - add evolutionary implications”

We expanded the section to discuss how dating TE insertions enables inference of evolutionary relationships among species and how our MSA-based approach can improve TE annotation in a way that complements current mainstream methods such as RepeatMasker. We also highlight how this framework provides an opportunity to study the tempo of TE-mediated genome innovation and offers potential insights into speciation.

[Additionally, the homoplasmy-free nature of TE provides unambiguous directionality, making them valuable markers for reconstructing phylogenetic relationships between species and among populations (100-103). With the increasing availability of MSAs across species, the construction of evolutionary relationships will become even more accessible and precise.

Harnessing MSA data, our computational tool TiMEstamp provides a rapid approach to estimate insertion times. By dating these events, we can assess the activity of various TEs across evolutionary timescales. Additionally, TiMEstamp can complement traditional sequence homology-based annotation methods, which can lead to ambiguous annotations. For example, non-Homo sapiens specific LINE-1 insertions have sometimes been mislabeled as L1HS, and certain L1M subfamilies are indeed primate-specific. By refining insertion timing, TiMEstamp can help improve existing TE annotations, making them more accurate and better aligned with evolutionary history. Over millions of years, TEs have significantly shaped our genome, though precisely how they have contributed to speciation and species-specific characteristics is only partially understood. Many primate-specific regulatory regions with cell type-specific and developmental activities are derived from TEs (104,105). Timing TE insertions by TiMEstamp relative to events in speciation could narrow down those involved in primate evolution and guide functional investigation.]

- “Identification of new RNA repeats forming chimeric insertion with LINE-1 - a short explanation or idea to explain why certain RNAs show specific patterns”

We added an explanatory note regarding the formation of these chimeric insertions, based on their structural features. We also expanded the section “Chimeric insertions with partial non-genetic RNAs” to further explain specific cases of chimeric LINE-1 insertions containing partial tRNA sequences.

[Structural features of chimeric RNA repeat/LINE-1 insertions provide clues about mechanisms underlying their formation. The sense orientation-bias relative to LINE-1 of U6, U3 and 28 rRNAs suggests the formation of these chimeras through RNA ligation or a template switch during the primary TPRT; in contrast, anti-sense orientation bias may indicate twin-priming as a mechanism.]

[Consistent with this, previous studies have demonstrated that tRNA fragments can selfprime ORF2p-mediated retrotransposition (110) and serve as primers for other retrotransposons in various organisms (133-137), highlighting their exceptional priming capacity. Additionally, their high abundance in cells (138) may enable them to exploit exposed DNA ends and initiate RT reactions.]

- “Chimeric insertions may have driven LINE-1 evolution - elaborate on mechanistic basis”

We elaborated on the potential mechanisms by which chimeric insertions may contribute to the evolution of LINE-1 promoters.

[Transcription driven by the internal RNAPolIII promoter of the LINE-1 5' UTR is the first

critical step in the LINE-1 life cycle. Successively active LINE-1 subfamilies often have distinctive 5' UTRs compared to LINE-1 predecessors with declining activity, whereas the ORF2 and ORF1 sequences remain relatively conserved among primate LINE-1 families. The acquisition of novel 5' UTRs has slowed only recently, following the emergence of L1PA8 (5). The punctuated evolution of the 5' UTR without transitional intermediate sequences suggests that LINE-1 may employ a distinctive mechanism to drive this diversification. We speculate that LINE-1 achieves these dramatic “swaps” of the 5' UTR through its ability to recombine with other RNAs and form chimeric insertions in the genome.]

Figures

- “Figure 2D heatmap difficult to interpret”

Thank you. We have revised the figure by adding phylogenetic labels and clarifying our description in the figure legend.

- “Figure 3F pie chart hard to interpret; increase font sizes in scatter plot legends”

Revised for clarity with larger fonts and an adjusted color scheme.

- “Figure 5 dense; suggest splitting into two figures”

We agree and have reorganized this figure, relocating elements to Supplementary Figure 7 to improve clarity and accessibility.

Reviewer #2

We thank Reviewer #2 for the encouraging comments and helpful suggestions.

- Explore the relationship TE driven eQTL, the regulatory role of the TEs and TEs driven exonization.

We thank the reviewer for highlighting this important clinical relevance of TE. The example provided by Reviewer 2, together with other reports, have demonstrated that TE driven eQTL in human populations (1) and in cancers (2); that exonization can be driven by SVA (3), Alu (4-7) and LINE-1 elements (8,9), and several studies have already established the clinical significance of such events (8,10-12). A recent study further showed that the incorporation of LINE-1 into splicing is normally suppressed by factors such as the SAFB complex, and knockout of this complex permits aberrant inclusion of LINE-1 sequences into mRNA. We speculate that disruption of these LINE-1 suppressive functions in disease could likewise enable TE-derived sequences to be aberrantly spliced, thereby contributing to pathogenesis. As our current study is focused on the identification and characterization of chimeric LINE-1 insertions (i.e., that have retrotransposed), we do not expand the scope of this work pursue a broader analysis of TE exonization in the transcriptome, though we agree that this remains a valuable direction for future research.

References:

1. Wang, L., Rishishwar, L., Marino-Ramirez, L. and Jordan, I.K. (2017) Human populationspecific gene expression and transcriptional network modification with polymorphic transposable elements. *Nucleic Acids Res*, 45, 2318-2328.
2. Lykoskoufis, N.M.R., Planet, E., Ongen, H., Trono, D. and Dermitzakis, E.T. (2024) Transposable elements mediate genetic effects altering the expression of nearby genes in colorectal cancer. *Nat Commun*, 15, 749.
3. Hancks, D.C., Ewing, A.D., Chen, J.E., Tokunaga, K. and Kazazian, H.H., Jr. (2009) Exon-trapping mediated by the human retrotransposon SVA. *Genome Res*, 19, 1983-1991.
4. Sorek, R., Lev-Maor, G., Reznik, M., Dagan, T., Belinky, F., Graur, D. and Ast, G. (2004) Minimal conditions for exonization of intronic sequences: 5' splice site formation in alu

exons. *Mol Cell*, 14, 221-231.

5. Sorek, R., Ast, G. and Graur, D. (2002) Alu-containing exons are alternatively spliced. *Genome Res*, 12, 1060-1067.

6. Krull, M., Brosius, J. and Schmitz, J. (2005) Alu-SINE exonization: en route to protein-coding function. *Mol Biol Evol*, 22, 1702-1711.

7. Shen, S., Lin, L., Cai, J.J., Jiang, P., Kenkel, E.J., Stroik, M.R., Sato, S., Davidson, B.L. and Xing, Y. (2011) Widespread establishment and regulatory impact of Alu exons in human genes. *Proc Natl Acad Sci U S A*, 108, 2837-2842.

8. Pasquesi, G.I.M., Allen, H., Ivancevic, A., Barbachano-Guerrero, A., Joyner, O., Guo, K., Simpson, D.M., Gapin, K., Horton, I., Nguyen, L.L. et al. (2024) Regulation of human interferon signaling by transposon exonization. *Cell*, 187, 7621-7636 e7619.

9. Arribas, Y.A., Baudon, B., Rotival, M., Suarez, G., Bonte, P.E., Casas, V., Roubert, A., Klein, P., Bonnin, E., McHich, B. et al. (2024) Transposable element exonization generates a reservoir of evolving and functional protein isoforms. *Cell*, 187, 7603-7620 e7622.

10. Taniguchi-Ikeda, M., Kobayashi, K., Kanagawa, M., Yu, C.C., Mori, K., Oda, T., Kuga, A., Kurahashi, H., Akman, H.O., DiMauro, S. et al. (2011) Pathogenic exon-trapping by SVA retrotransposon and rescue in Fukuyama muscular dystrophy. *Nature*, 478, 127-131.

11. Vorechovsky, I. (2010) Transposable elements in disease-associated cryptic exons. *Hum Genet*, 127, 135-154.

12. Goncalves, A., Oliveira, J., Coelho, T., Taipa, R., Melo-Pires, M., Sousa, M. and Santos, R. (2017) Exonization of an Intronic LINE-1 Element Causing Becker Muscular

Dystrophy as a Novel Mutational Mechanism in Dystrophin Gene. *Genes (Basel)*, 8.

Referees' report, second round of review:

Authors' response to the second round of review